# DIRECTION-MAGNITUDE DECOUPLING FOR FAST VIDEO GENERATION WITH FLOW MATCHING MODELS

## ABSTRACT

Flow matching models for video generation achieve impressive performance but suffer from high computational overhead due to iterative denoising. In fact, the original model is not necessary for all denoising steps, allowing some steps to use lightweight alternatives for faster processing. However, directly using caching or lightweight models can deviate from the original denoising trajectory, resulting in suboptimal performance. Through empirical analysis, we find that lightweight models can robustly capture the magnitude components of the original model's output, while caching provides reliable directional guidance. Building on this insight, we propose the Direction-Magnitude Decoupling (DMD) method, which adaptively employs a direction-calibrated lightweight model as a substitute for the original model to accelerate inference and effectively correct deviations in the denoising trajectory. Moreover, DMD further reduces inference costs by reusing magnitude information under classifier-free guidance (CFG). As a result, DMD offers a more reliable and lightweight solution to accelerate denoising. Experiments show that DMD outperforms existing acceleration methods, delivering significant speedups (e.g., up to 2.95× on Wan2.1) while maintaining visual fidelity.

## 1 INTRODUCTION

Driven by diffusion models (Dhariwal & Nichol, 2021; Ho et al., 2020), visual generation has achieved remarkable success in recent years. A growing body of research (Wang et al., 2025; Zheng et al., 2024; Yang et al., 2025; Xu et al., 2024; Ma et al., 2025a) continues to explore the frontiers of diffusion models, achieving impressive levels of fidelity and temporal coherence in video generation. Notably, flow matching (Lipman et al., 2023) has emerged as a compelling framework for achieving faster convergence and improved controllability in video generation (Esser et al., 2024; Liu et al., 2023), and it is increasingly being adopted by state-of-the-art (SOTA) methods.

However, the prohibitive inference latency of diffusion models remains a critical bottleneck (Li et al., 2023b). This core limitation arises from the inherently sequential denoising process, which becomes even more pronounced as models scale to higher resolutions and longer video durations (Chen et al., 2024b). Existing work based on distillation (Salimans & Ho, 2022; Meng et al., 2023) or post-training quantization (Chen et al., 2025) offers potential acceleration but requires costly model training and additional data resources. In contrast, training-free methods, including cache-based methods (Ma et al., 2024; Liu et al., 2025a) and large-small model collaborative inference method (Cheng et al., 2025), recognize that the original model is not required for all denoising steps and instead employ lightweight substitutes for certain steps to accelerate inference. These methods are easy to use, cost-effective, and generally applicable, making them the main focus of this work.

Cache-based methods (Ma et al., 2024; Chen et al., 2024c) observe that model outputs are similar across consecutive timesteps during denoising and propose to reduce redundancy using caching mechanisms, such as residual reuse (Ma et al., 2025b). Although caching captures this redundancy, its static nature can cause rapid error accumulation and degrade performance under high cache reuse rates (example in Figure 1, Teacache (Liu et al., 2025b)). The large-small model collaborative inference method SRDiffusion (Cheng et al., 2025) posits that the original large model plays a critical role during the initial stages of semantic construction. In the subsequent stages, a small model from the same family can be employed to refine visual details, serving as a lightweight alternative that accelerates the process. Despite the promising acceleration benefits, directly using the bias output of

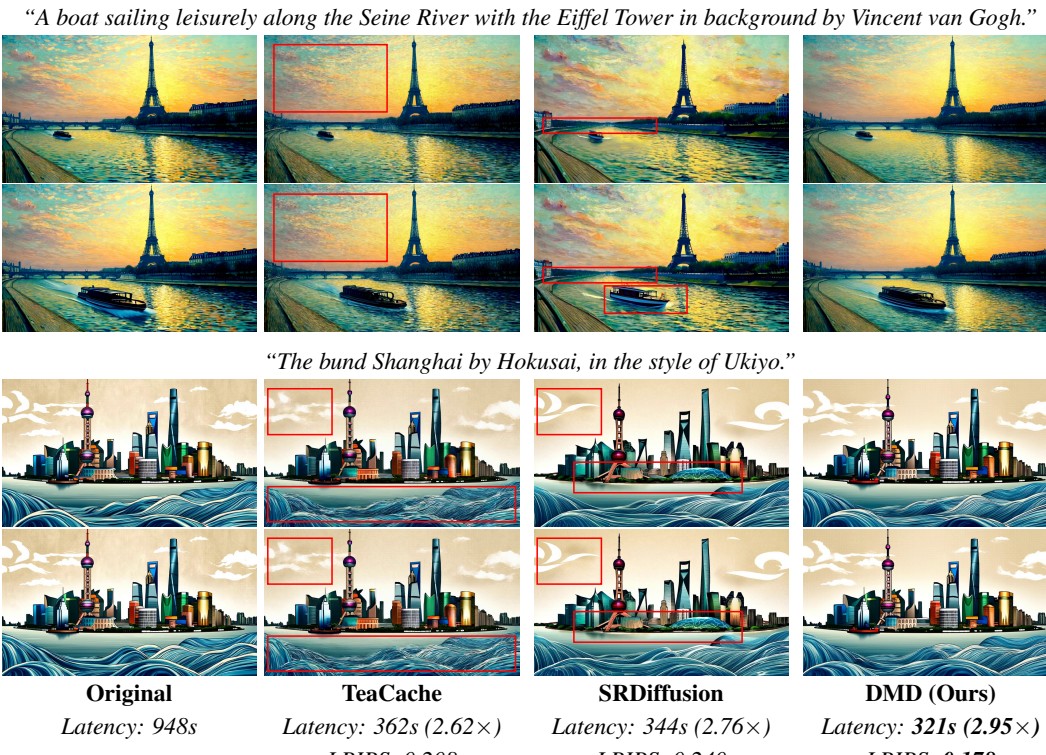

*"A boat sailing leisurely along the Seine River with the Eiffel Tower in background by Vincent van Gogh."*

*"The bund Shanghai by Hokusai, in the style of Ukiyo."*

| **Original** | **TeaCache** | **SRDiffusion** | **DMD (Ours)** |
|---|---|---|---|
| *Latency: 948s* | *Latency: 362s (2.62×)* | *Latency: 344s (2.76×)* | *Latency: **321s (2.95×)*** |
| | *LPIPS: 0.208* | *LPIPS: 0.240* | *LPIPS: **0.178*** |

Figure 1: Comparison of visual fidelity and inference speed with competing methods. Latency is measured on a single A100 GPU. LPIPS denotes learned perceptual image patch similarity. Video synthesis configuration: 81 frames, 5s, 480P on Wan2.1-14B. Our method consistently produces outputs that better preserve the original video content while delivering improved acceleration.

a limited-capacity small model may lead to denoising trajectory deviations, resulting in suboptimal visual retention (example in Figure 1, SRDiffusion (Cheng et al., 2025)).

To address this issue, we conduct an error source analysis under the flow matching framework, comparing cache-based residual reuse outputs and small model outputs with those of the original large model. By decoupling the output into magnitude and directional components, our empirical analysis shows that errors in residual-reuse outputs primarily arise from magnitude discrepancies, while the directional component is well approximated. In contrast, errors in small model outputs are mainly due to directional misalignment, whereas the magnitude information is estimated reliably.

Based on these insights, we propose a novel method called DMD, which accelerates inference by adaptively employing a reliable lightweight alternative to the original large model. Specifically, DMD improves the outputs of lightweight models by calibrating them along directions obtained from residual reuse, enabling the lightweight alternative output to better approximate the original denoising trajectory. To prevent the accumulation of directional errors from invariant residual reuse, DMD adaptively switches to a larger model for directional recalibration when it detects excessive error growth. Moreover, DMD further reduces inference costs by reusing magnitude information under CFG. As a result, DMD can preserve the content of the original video more faithfully while delivering improved acceleration (example in Figure 1, DMD). Qualitative experimental comparisons demonstrate that our DMD outperforms existing acceleration strategies in both inference speed and video generation fidelity, verifying the effectiveness of our proposal.

In summary, our contributions are as follows:

- We empirically find that, under the flow matching framework, the small model can robustly estimate the output magnitude component of the original large model from the same family, while reusing cached residuals can reliably approximate the directional component.

- We propose DMD, which adaptively employs a direction-calibrated small model as a lightweight alternative to the original large model during inference, while maintaining a faithful approximation of the original denoising trajectory. It also introduces CFG reuse to further accelerate inference.

- We evaluate DMD on various flow-based video generation models and demonstrate that our approach consistently outperforms existing acceleration strategies, achieving greater speedups while simultaneously delivering superior visual fidelity.

## 2  RELATED WORK

**Diffusion and Flow-based Models.** In generative modeling, diffusion models (Ho et al., 2020; Sohl-Dickstein et al., 2015) have become foundational for their ability to produce high-quality, diverse outputs (Blattmann et al., 2023; Chen et al., 2024a). Early methods, such as DDPM (Ho et al., 2020), DDIM (Song et al., 2021), and EDM (Karras et al., 2022), are score-based models that learn the stochastic differential equations (SDEs) governing the diffusion process. In contrast, flow matching (Lipman et al., 2023) provides an alternative by using ordinary differential equations (ODEs) to model sample trajectories, offering more stable and efficient generation through a deterministic mapping from the prior to the target distribution. Numerous studies (Liu et al., 2023; Esser et al., 2024; Kong et al., 2024) have shown that flow matching models converge faster and offer better controllability in video generation, making them a strong alternative to stochastic diffusion models with better interpretability and stability. Thus, our analysis is based on flow matching models, aiming to provide a lightweight alternative that enables more efficient inference.

**Diffusion Model Acceleration.** Due to their high inference costs, diffusion models have prompted extensive efforts for acceleration. One line of research focuses on methods based on training or fine-tuning, such as post-training quantization (Shang et al., 2023; He et al., 2023; Li et al., 2023a; Wang et al., 2024; Chen et al., 2025), reducing sampling steps through progressive model distillation (Salimans & Ho, 2022; Meng et al., 2023; Lin & Yang, 2024; Sauer et al., 2024), or employing consistency models (Song et al., 2023; Luo et al., 2023). However, they necessitate costly retraining and additional data resources, which can constrain their feasibility for broad implementation.

Another line of research focuses on training-free methods. Foundational methods like DDIM (Song et al., 2021) enable fewer sampling steps without sacrificing quality. Additional studies use efficient ODE or SDE solvers (Song & Ermon, 2019; Karras et al., 2022; Lu et al., 2022), employing pseudo-numerical methods for faster sampling. Furthermore, a series of studies (Ma et al., 2024; Chen et al., 2024c; Yu et al., 2025) have shown that not every step in the iterative denoising process requires the full original model, motivating various lightweight alternatives to accelerate inference. Cache-based methods (Wimbauer et al., 2024; Liu et al., 2025c; Ma et al., 2025b) exploit redundancies in model outputs across consecutive timesteps, caching outputs according to criteria such as content complexity (AdaCache (Kahatapitiya et al., 2024)) or timestep embeddings (Teacache (Liu et al., 2025b)) to reduce computational overhead. Large-small model collaborative inference strategies (Yang et al., 2024) leverage lightweight models from the same family as substitutes to accelerate inference. For example, SRDiffusion (Cheng et al., 2025) uses the original large model during the early stages of denoising to generate coarse semantic structures, which are then refined by a lightweight small model responsible for producing fine visual details. However, the limited capacity of the small model can lead to deviations from the original large model's denoising trajectory, ultimately compromising visual retention.

## 3  PRELIMINARIES

**Flow Matching** (Lipman et al., 2023) is a family of generative models that transport samples from a data distribution $p_0(x)$ into a simple prior distribution $p_1(x)$ (e.g., Gaussian). The probability path $p_t(x)$ is constructed to interpolate between $p_0(x)$ and $p_1(x)$ over the continuous time variable $t \in [0, 1]$, and is commonly defined using linear interpolation (Esser et al., 2024):

$$p_t(x) = (1 - t) \cdot p_0(x) + t \cdot p_1(x).$$

The flow matching model is trained by learning a time-dependent velocity field $\frac{d}{dt}x_t = v_\theta(x_t, t)$, where $v_\theta(x_t, t)$ is parameterized by a neural network with parameters $\theta$. Formally, the model is

optimized by minimizing the following loss function:

$$\mathcal{L}_{\text{FM}}(\theta) = \mathbb{E}_{t, x_0 \sim p_0(x), x_1 \sim p_1(x)} \left[ v_\theta(x_t, t) - (x_1 - x_0) \right].$$

At generation time, new samples can be generated using any ODE solver (Süli & Mayers, 2003).

**Residual Reuse.** A series of studies (Kahatapitiya et al., 2024; Liu et al., 2025b) have consistently observed that the denoising process often involves redundant computations, suggesting that reuse strategies could be employed to skip certain steps. Specifically, we define the residual $r$ at timestep $t$ as the difference between the model's output and the corresponding input:

$$r = v_\theta(x_t, t) - x_t. \tag{1}$$

In subsequent steps, this residual can be reused by adding it to the input, thus directly estimating the residual reuse output (e.g., $\hat{v}_\theta(x_{t-1}, t-1) = x_{t-1} + r$). Although $r$ captures the update signal, repeated reuse of the invariant residual can lead to error accumulation, ultimately degrading the quality of the visual output.

**Classifier-Free Guidance (CFG)** (Ho & Salimans, 2022) is a widely adopted strategy for improving the quality of conditional generation by steering samples toward the specified input condition. Let $c$ denote the input condition, such as a class label or a text prompt. In CFG, a single flow model $v_\theta(x_t, t \mid c)$ is trained to output both conditional and unconditional velocity fields. During sampling, the model performs two forward evaluations: one conditioned on $c$ and the other unconditioned, with $c = \emptyset$. The guided velocity field is formed by:

$$v_\theta^{\text{cfg}}(x_t, t \mid c) = v_\theta(x_t, t \mid c = \emptyset) + w \cdot \Big( v_\theta(x_t, t \mid c) - v_\theta(x_t, t \mid c = \emptyset) \Big), \tag{2}$$

where $w$ denotes the guidance scale. Setting $w = 1$ represents the non-guided case.

## 4 PROPOSED METHOD

### 4.1 EMPIRICAL ANALYSIS

Recent flow-based model families (Wang et al., 2025; Xu et al., 2024) offer both large and small model variants. By sharing a unified VAE within these model families, the models enable seamless switching between the high-capacity large model ($\theta$) and the lightweight small model ($\varphi$) during sampling, thus opening up opportunities for large-small model collaboration to accelerate inference.

Let $v_\theta(x_t, t)$ and $v_\varphi(x_t, t)$ denote the outputs of the large and small models, respectively, and $\hat{v}_\theta(x_t, t)$ represents the residual-reused outputs from the original large model. In this paper, we explore how $v_\varphi(x_t, t)$ and $\hat{v}_\theta(x_t, t)$ relate to $v_\theta(x_t, t)$, aiming to identify better lightweight alternatives for replacing the costly inference of the original large model. Specifically, we decouple the model outputs into direction and magnitude components for separate analysis. As shown in Figures 2a, 2b, 2d, and 2e, based on the empirical analysis of the intermediate diffusion process (i.e., from 20% to 95%), we derive two key observations:

***Observation 1:*** *In terms of directional component, the residual-reuse outputs $\hat{v}_\theta(x_t, t)$ are closely aligned with those of the original large model $v_\theta(x_t, t)$, i.e., $sim(\hat{v}_\theta(x_t, t), v_\theta(x_t, t)) \approx 1$, where $sim(\cdot, \cdot)$ denotes the cosine similarity. In contrast, the small model's directional estimates typically diverge more, i.e., $sim(v_\theta(x_t, t), \hat{v}_\theta(x_t, t)) > sim(v_\theta(x_t, t), v_\varphi(x_t, t))$.*

***Observation 2:*** *In terms of magnitude component, the small-model outputs $v_\varphi(x_t, t)$ closely resemble those of the large model $v_\theta(x_t, t)$, i.e., $\frac{\|v_\theta(x_t, t)\|_2}{\|v_\varphi(x_t, t)\|_2} \approx 1$, where $\| \cdot \|_2$ denotes the $\ell_2$ norm. In contrast, the residual-reuse outputs typically show a larger magnitude discrepancy, i.e., $\left| \frac{\|v_\theta(x_t, t)\|_2}{\|v_\varphi(x_t, t)\|_2} - 1 \right| < \left| \frac{\|v_\theta(x_t, t)\|_2}{\|\hat{v}_\theta(x_t, t)\|_2} - 1 \right|.$*

With respect to the direction component, the residual-reuse outputs closely approximate those of the large model, whereas the small-model outputs display directional deviations. This implies that employing the small model directly for collaborative inference can lead to divergence from the large model's denoising trajectory, resulting in suboptimal visual retention. Conversely, the linear path learning objective in flow matching models allows for efficient capture of directional information through residual reuse, a finding also supported by (Ma et al., 2025b).

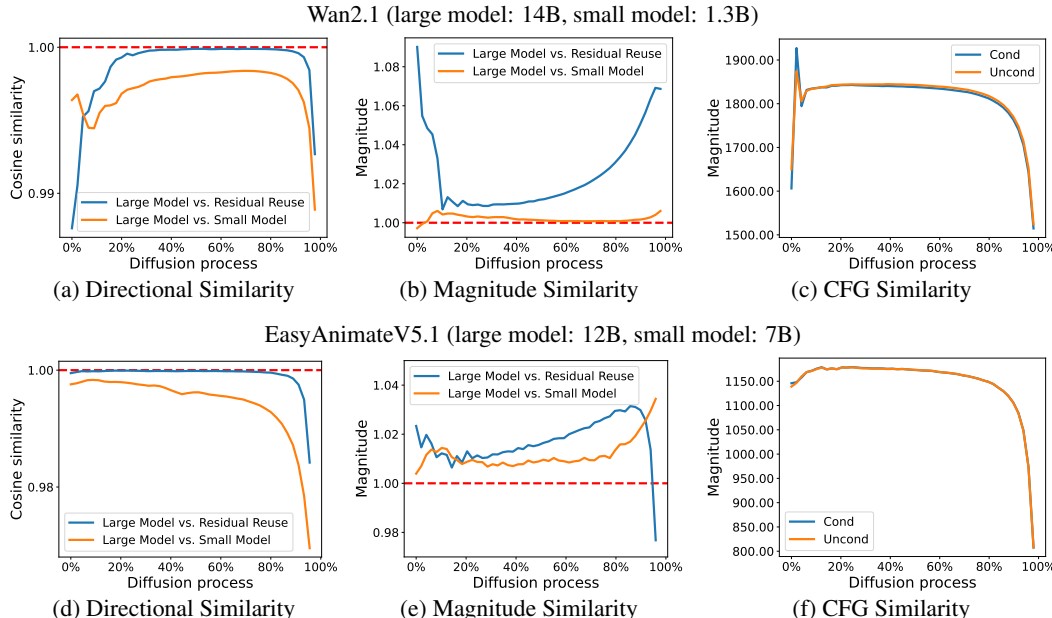

Figure 2: Visualization of decoupled model output relationships. (a,d) directional similarities of $v_\theta(x_t, t)$ with $v_\varphi(x_t, t)$ and $\hat{v}_\theta(x_t, t)$; (b,e) magnitude ratios of $v_\theta(x_t, t)$ to $v_\varphi(x_t, t)$ and to $\hat{v}_\theta(x_t, t)$; (c,f) magnitudes of $v_\varphi(x_t, t \mid c)$ (Cond) and $v_\varphi(x_t, t \mid c = \emptyset)$ (Uncond). Directional similarities are evaluated using cosine similarity, and magnitudes are measured using $\ell_2$ norms.

With respect to magnitude component, the outputs of the large and small models are comparatively consistent. This suggests that the small model can serve as an effective lightweight proxy for the large model in estimating output magnitude. However, the consecutive use of residual-reuse outputs with inconsistent magnitudes can lead to rapid error accumulation, thereby impairing performance.

Collectively, in approximating the outputs of the large model, the residual-reuse outputs provide more accurate directional estimates, whereas the small-model outputs better capture the output magnitudes. This insight informs the design of our subsequent acceleration strategy.

## 4.2 Acceleration Strategy

In this subsection, building upon the previous empirical analysis, we propose a novel inference acceleration strategy called DMD. The objective is to provide a reliable and lightweight alternative to the original large model. The framework of our proposal and its comparison with competing methods are illustrated in Figure 3, with a detailed description provided below.

**Lightweight Alternative Strategy.** As discussed in Section 4.1, $\hat{v}_\theta(x_t, t)$ and $v_\varphi(x_t, t)$ are, respectively, more accurate in capturing the direction and magnitude components of the original large model output $v_\theta(x_t, t)$. By leveraging this complementary advantage, DMD retains the magnitude from the lightweight model $v_\varphi(x_t, t)$ while utilizing the residual-reused output $\hat{v}_\theta(x_t, t)$ to guide the direction. Formally, at timestep $t$ with input $x_t$ and condition $c$, the output estimated by our DMD strategy can be expressed as:

$$v_{\text{DMD}}(x_t, t \mid c) = \underbrace{\|v_\varphi(x_t, t \mid c)\|_2}_{\text{Magnitude}} \cdot \underbrace{\frac{\hat{v}_\theta(x_t, t \mid c)}{\|\hat{v}_\theta(x_t, t \mid c)\|_2}}_{\text{Direction}} . \tag{3}$$

Compared to SRDiffusion (Cheng et al., 2025), which performs inference directly using a small model with directional prediction biases, DMD improves by calibrating the direction through residual reuse. As a result, DMD more closely approximates the denoising trajectory of the original large model, thus enhancing visual retention. Compared with Teacache (Liu et al., 2025b), a cache-based

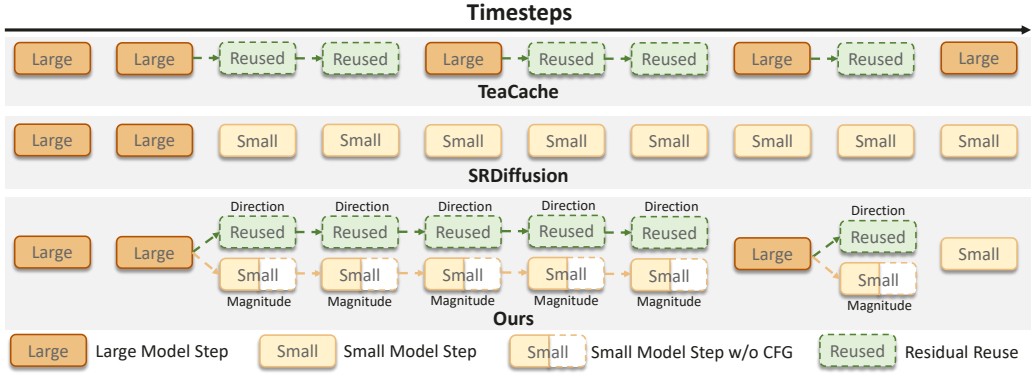

Figure 3: Comparison of our method with the adaptive caching method, TeaCache (Liu et al., 2025b), and the large-small model collaborative inference strategy, SRDiffusion (Cheng et al., 2025). Our method improves the approximation of the original large output by combining direction estimation from residual reuse with magnitude estimation from the small model, offering a more reliable lightweight alternative for accelerating inference.

method that reuses invariant residuals, DMD combined with the lightweight assistance of the small model for magnitude prediction. From the perspective of caching, DMD equips residual reuse with magnitude-aware capabilities, thereby improving visual fidelity.

**CFG Reuse.** As illustrated in Figures 2c and 2e, the magnitude of the model outputs in CFG are nearly identical for both the conditional output $v_\varphi(x_t, t \mid c)$ and the unconditional output $v_\varphi(x_t, t \mid c = \emptyset)$, i.e., $\|v_\varphi(x_t, t \mid c)\|_2 \approx \|v_\varphi(x_t, t \mid c = \emptyset)\|_2$. Consequently, a single forward evaluation to compute the magnitude of the conditional output in CFG is sufficient. This magnitude information can then be combined with both the residual-guided conditional and unconditional directions, further halving the inference cost of the small model. Formally, the unconditional output estimated by our DMD is expressed as:

$$v_{\text{DMD}}(x_t, t \mid c = \emptyset) = \underbrace{\|v_\varphi(x_t, t \mid c)\|_2}_{\text{Reuse}} \cdot \frac{\hat{v}_\theta(x_t, t \mid c = \emptyset)}{\|\hat{v}_\theta(x_t, t \mid c = \emptyset)\|_2}. \tag{4}$$

**Adaptive Strategy.** The reuse of invariant residuals in direction estimation lacks the inherent self-correction capacity of small-model magnitude estimation. Consequently, even minor directional errors can gradually accumulate over successive reuse steps, resulting in significant error amplification. Once the directional error grows beyond a tolerable threshold, it becomes necessary to invoke the original large model to correct the direction in a timely manner. In practice, we use the cumulative cosine error of the output directional components as the criterion for determining whether DMD should be applied. To mitigate the computational overhead of performing expensive forward evaluations with the large model, we instead leverage the small model as a surrogate to estimate the directional error. Formally, we maintain a running cumulative error $\mathcal{E}$ defined as:

$$\mathcal{E} = \sum_{i=t}^{t'-1} \Big(1 - sim(v_\varphi(x_i, i \mid c), v_\varphi(x_{t'-1}, t' - 1 \mid c))\Big), \tag{5}$$

where $t'$ denotes the most recent timestep at which a residual is cached. $\mathcal{E}$ quantifies the accumulated deviation of output directions over successive timesteps: smaller values indicate that the directions remain well-aligned and can be safely reused, whereas larger values trigger the use of the original large model to update the directions. Accordingly, the decision function at timestep $t$ is defined as:

$$D_\tau(t) = \begin{cases} v_{\text{DMD}}(x_t, t \mid c), & \text{if } \mathcal{E} \leq \tau; \\ v_\theta(x_t, t \mid c), & \text{if } \mathcal{E} > \tau, \end{cases} \tag{6}$$

where $\tau$ denotes the threshold for the maximum tolerable directional error, serving to balance computational efficiency and visual quality. The pseudo code of DMD is available in Algorithm 1.

---

**Algorithm 1** Direction-Magnitude Decoupling (DMD)

---

1: **Initialize** latent variable $x_T$, DMD interval $[T_1, T_2]$, directional error $\mathcal{E}$, and threshold $\tau$
2: **for** each timestep $t$ in $\{T_2, \ldots, T_1\}$ **do**
3:  **if** $\mathcal{E} > \tau$ **then**
4:      Predict outputs: $v_t \leftarrow \text{LargeModel}(x_t, t)$         # With CFG
5:      Cache residuals: $r \leftarrow v_t - x_t$
6:      Reset the error: $\mathcal{E} \leftarrow 0$
7:  **else**
8:      Predict output: $v'_t \leftarrow \text{SmallModel}(x_t, t)$         # Without CFG
9:      Reuse residuals: $\hat{v}_t \leftarrow x_t + r$
10:      DMD: $v_t \leftarrow \text{norm}(v'_t) \cdot (\hat{v}_t \,/\, \text{norm}(\hat{v}_t))$
11:      Update the error: $\mathcal{E} \leftarrow \mathcal{E} +$ directional error of $v'_t$
12:  **end if**
13:  Update latents: $x_{t-1} \leftarrow \text{ODEStep}(v_t, t, x_t)$
14: **end for**
15: **return** $x_{T_1}$

---

## 5 EXPERIMENTS

In this section, we first describe our experimental setup, then present the quantitative results comparison, followed by the ablation analysis.

### 5.1 EXPERIMENTAL SETUP

**Base Models and Baselines.** To verify the effectiveness of our method, we compare it against the most competitive methods across different flow matching models. Specifically, we apply our acceleration method to Wan2.1 (Wang et al., 2025) and EasyAnimateV5.1 (Xu et al., 2024), each offered in two model scales with a shared VAE. Wan2.1 offers 14B and 1.3B variants, while EasyAnimateV5.1 provides 12B and 7B variants. We compare our method with two competitive baselines: the large-small model collaborative inference method, SRDiffusion (Cheng et al., 2025), and the cache-based method, TeaCache (Liu et al., 2025b).

**Evaluation Metrics.** For the quantitative evaluation of video generation acceleration methods, we focus on two key aspects: inference efficiency and visual quality. Following (Cheng et al., 2025; Liu et al., 2025b), inference efficiency is assessed via per-sample inference latency, whereas visual quality is evaluated using VBench (Huang et al., 2024) in conjunction with three widely adopted metrics capturing perceptual consistency, pixel-level fidelity, and structural similarity: Learned Perceptual Image Patch Similarity (LPIPS) (Zhang et al., 2018), Peak Signal-to-Noise Ratio (PSNR), and Structural Similarity Index Measure (SSIM) (Wang et al., 2004).

**Implementation Detail.** Following the evaluation setup in (Liu et al., 2025b), we use the standard prompt set provided by VBench (Huang et al., 2024) for text-conditioned video generation. All experiments are conducted on NVIDIA A100 GPUs, with FlashAttention (Dao et al., 2022) enabled by default. Latency is measured on a single A100 GPU. Following (Xia et al., 2025; Ma et al., 2025b), we preserve the first 20% of the diffusion steps with the large model for TeaCache and our DMD, as these initial steps are critical to the overall generation process. In line with (Liu et al., 2025b), the error accumulation threshold for the TeaCache baseline is set to 0.2. According to (Cheng et al., 2025), the threshold for transitioning from the large model to the small model in the SRDiffusion baseline is set to 0.1. For our DMD, we set the cumulative directional error threshold $\tau$ in Eq. (6) to 0.005. Furthermore, we employ the small model for the final 5% of the steps to ensure the reliability of visual outputs.

### 5.2 MAIN RESULTS

**Quantitative Comparison.** Table 1 provides a comprehensive quantitative evaluation of our method compared to the most competitive baselines. The results indicate that: (1) our DMD consistently achieves highest acceleration (2.95× for Wan2.1, 1.90× for EasyAnimateV5.1) while preserving high visual retention (e.g., LPIPS 0.178, PSNR 22.72, SSIM 0.748 on Wan2.1), demonstrating the effectiveness of our lightweight alternative strategy; (2) compared with the large-small model

Table 1: Quantitative evaluation of inference efficiency and visual quality in video generation models. Visual retention metrics, including LPIPS, SSIM, and PSNR, are calculated against the original large models (Wan2.1-14B and EasyAnimateV5.1-12B).

| Method | Efficiency | | Visual Quality | | | |
|---|---|---|---|---|---|---|
| | Speedup↑ | Latency↓ | LPIPS↓ | PSNR↑ | SSIM↑ | VBench↑ |
| **Wan2.1** (832×480, 81 frames, $T = 50$) | | | | | | |
| Wan2.1-14B | 1× | 948 s | - | - | - | 83.09% |
| Wan2.1-1.3B | - | 192 s | 0.597 | 11.81 | 0.336 | 81.00% |
| TeaCache (Liu et al., 2025b) | 2.62× | 362 s | 0.208 | 20.95 | 0.680 | 82.41% |
| SRDiffusion (Cheng et al., 2025) | 2.75× | 344 s | 0.240 | 19.10 | 0.636 | 83.01% |
| **DMD (Ours)** | **2.95×** | **321 s** | **0.178** | **22.72** | **0.748** | 82.62% |
| **EasyAnimateV5.1** (672×384, 49 frames, $T = 50$) | | | | | | |
| EasyAnimateV5.1-12B | 1× | 246 s | - | - | - | 78.89% |
| EasyAnimateV5.1-7B | - | 133 s | 0.622 | 12.90 | 0.404 | 75.71% |
| TeaCache (Liu et al., 2025b) | 1.84× | 134 s | 0.179 | 21.30 | 0.708 | 78.53% |
| SRDiffusion (Cheng et al., 2025) | 1.57× | 157 s | 0.380 | 17.75 | 0.540 | 78.24% |
| **DMD (Ours)** | **1.90×** | **129 s** | **0.150** | **22.66** | **0.755** | 78.65% |

collaborative inference baseline SRDiffusion, DMD leverages direction-calibrated outputs from a lightweight model to more faithfully approximate of the original denoising trajectory, rather than directly using the biased directional predictions of the small model for acceleration. This design, further enhanced by CFG reuse, yields higher acceleration while delivering superior visual fidelity; and (3) compared with the cache-based baseline TeaCache, DMD can be considered as adopting a magnitude-aware residual reuse strategy assisted by the lightweight model, rather than relying on invariant residual reuse. Consequently, our method can achieve higher reuse rates through more reliable lightweight substitutions, enabling faster acceleration while maintaining better visual fidelity.

**Visualization Results.** Figure 1 and Appendix F showcase a comparison of video generation results from the original Wan2.1-14B model, TeaCache (Liu et al., 2025b), SRDiffusion (Cheng et al., 2025), and our DMD. The results show that: (1) in pursuit of higher acceleration, TeaCache accumulates excessive errors due to residual reuse across multiple steps, ultimately degrading the quality of the generated videos; (2) SRDiffusion relies on a less capable lightweight model for content rendering, which can deviate from the denoising trajectory of the original large model, leading to the loss of content details and suboptimal visual retention; and (3) our method more faithfully preserves the denoising trajectory of the original large model, achieving higher fidelity in the generated details while delivering greater acceleration.

## 5.3 ABLATION STUDIES

**The Effect of Threshold $\tau$.** Table 2 examines the impact of varying the threshold $\tau$ in Eq. (6) on inference efficiency and visual quality. The results indicate that: (1) A higher $\tau$ allows greater accumulation of directional error, enabling the lightweight model to perform more substitution steps and thus increasing acceleration. However, excessive error may adversely affect performance. For example, setting $\tau = 0.010$ yields a 3.12× speedup but reduces fidelity (LPIPS = 0.207, PSNR = 22.21, SSIM = 0.728). (2) A lower $\tau$ limits error accumulation, facilitating timely corrections by the original large model, which reduces efficiency but improves quality. For example, $\tau = 0.001$ results in LPIPS = 0.131, PSNR = 23.76, SSIM = 0.786, with a reduced speedup of 2.21×. (3) An intermediate value, $\tau = 0.005$, provides a favorable trade-off, achieving a 2.95× speedup while maintaining acceptable visual fidelity (LPIPS = 0.178, PSNR = 22.72, SSIM = 0.748).

**The Effect of CFG Reuse.** Table 3 compares the effect of reusing the magnitude of the conditional output for the unconditional output in CFG. The results demonstrate that: (1) in terms of inference latency, reusing the magnitude of the conditional output for the unconditional output can reduce the inference cost of small models by half, providing an additional 1.20× speedup in overall inference and further decreasing computational overhead; and (2) in terms of visual quality, since

Table 2: Ablation study of threshold $\tau$ in Eq. (6) on Wan2.1. A larger threshold enables a greater proportion of steps to be approximated by the lightweight small-model, yielding higher acceleration at the expense of visual quality.

| Threshold $\tau$ | Efficiency | | Visual Quality | | | |
|---|---|---|---|---|---|---|
| | Speedup↑ | Latency↓ | LPIPS↓ | PSNR↑ | SSIM↑ | VBench↑ |
| Wan2.1-14B | 1× | 948 s | - | - | - | 83.09% |
| $\tau = 0.001$ | 2.21× | 428 s | 0.131 | 23.76 | 0.786 | 82.74% |
| $\tau = 0.002$ | 2.76× | 344 s | 0.148 | 23.33 | 0.771 | 82.85% |
| $\tau = 0.005$ | 2.95× | 321 s | 0.178 | 22.72 | 0.748 | 82.62% |
| $\tau = 0.010$ | 3.12× | 303 s | 0.207 | 22.21 | 0.728 | 82.20% |

Table 3: Ablation study of CFG Reuse on Wan2.1. Visual retention metrics (LPIPS, SSIM, and PSNR) are computed relative to the Wan2.1-14B model, with $\tau$ set to 0.005.

| CFG Reuse | Efficiency | | Visual Quality | | | |
|---|---|---|---|---|---|---|
| | Speedup↑ | Latency↓ | LPIPS↓ | SSIM↑ | PSNR↑ | VBench↑ |
| × | - | 385 s | 0.177 | 22.73 | 0.750 | 82.64% |
| ✓ | 1.20× | 321 s | 0.178 | 22.72 | 0.748 | 82.62% |

the magnitudes of the conditional and unconditional outputs in CFG are nearly identical, reusing the magnitude effectively preserves the original visual quality. Quantitative evaluations demonstrate that the performance exhibits a negligible decline after CFG reuse, further confirming the effectiveness of this acceleration strategy.

**Acceleration in Multi-GPU Setups.** Consistent with prior work (Wang et al., 2025) that employs context parallel strategies for multi-GPU video generation, we evaluate DMD under this setting. Figure 4 presents results on Wan2.1, comparing the latency of DMD with SRDiffusion (Cheng et al., 2025) and TeaCache (Liu et al., 2025b) on A100 GPUs. Experimental results show that, when scaled across multiple GPUs, DMD consistently achieves higher inference efficiency than both SRDiffusion and TeaCache, highlighting the effectiveness of the DMD inference strategy in multi-GPU settings.

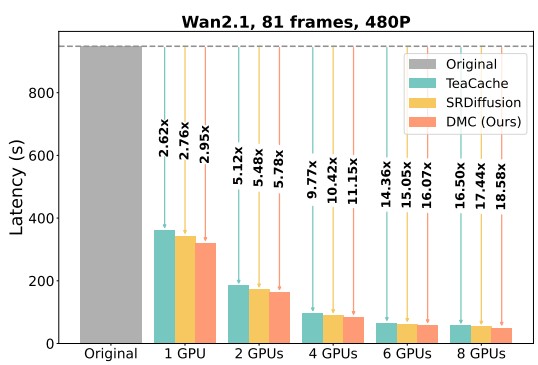

Figure 4: Multi-GPU inference efficiency analysis.

## 6 CONCLUSION

This study explores the use of lightweight alternatives to accelerate flow-based video generation. Our empirical analysis indicates that, within the same model family, the small model can effectively capture the magnitude information of the original large model, while the residual reuse output provides reliable directional guidance. Building on this, we propose DMD, a training-free method that accelerates inference by replacing part of the denoising steps with reliable lightweight alternatives. Specifically, DMD adaptively combines magnitude estimates from the small model with residual-based directional guidance to approximate the denoising trajectory of the original large model. Moreover, DMD further reduces inference costs through CFG reuse. Extensive experimental results demonstrate that DMD not only achieves substantial acceleration but also preserves high visual fidelity. We believe our work offers a novel perspective and an effective practical solution for accelerating flow matching models, facilitating their broader adoption.

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

## A   ANALYSIS OF DMD IN EARLY DENOISING

In the early denoising stage, the model needs to synthesize semantic content from pure noise (Cheng et al., 2025). Unlike the intermediate stage (20%–95%), the initial phase is critical and exhibits low redundancy, as confirmed by prior post-hoc acceleration studies (Cheng et al., 2025; Liu et al., 2025b; Lv et al., 2025). Consequently, the directional component, which relies on redundancy-based residual reuse, is constrained at this point. In contrast, magnitude estimation remains relatively stable, as it is derived from lightweight neural networks. Quantitative results demonstrating the effects of applying lightweight substitutions in the early denoising stage are presented in Table 6. The results show that lightweight substitutions in these critical early steps incur substantial performance degradation. Overall, it is generally advisable to rely on the original large model for reliable generation during the early denoising phase, a strategy that has also been widely adopted in post-hoc acceleration methods (Cheng et al., 2025; Liu et al., 2025b; Lv et al., 2025).

Table 4: Empirical analysis of lightweight substitution intervals. The threshold $\tau$ is set to 0.005.

| Interval | Efficiency | | Visual Quality | | | |
|---|---|---|---|---|---|---|
| | Speedup↑ | Latency↓ | LPIPS↓ | SSIM↑ | PSNR↑ | VBench↑ |
| 0%–95% | 3.85× | 246 s | 0.567 | 13.43 | 0.431 | 79.02% |
| 20%–95% (Ours) | 2.95× | 321 s | 0.178 | 22.72 | 0.748 | 82.62% |

## B   COMBINING DMD WITH EFFICIENT ODE SOLVERS

Efficient ODE solvers aim to achieve high-quality sampling with fewer steps, whereas our method reduces inference cost by replacing parts of the original large-model inference process with lightweight substitutes. Table 5 presents a supplementary analysis of step reduction using DMD with the UniPC (Zhao et al., 2023) and DPM++ (Lu et al., 2025) solvers, demonstrating its compatibility with efficient ODE solvers.

Table 5: Empirical analysis of lightweight substitution intervals. The threshold $\tau$ is set to 0.005.

| Methods | ODE solver | Efficiency | | Visual Quality | | | |
|---|---|---|---|---|---|---|---|
| | | Speedup↑ | Latency↓ | LPIPS↓ | SSIM↑ | PSNR↑ | VBench↑ |
| Wan-14B (T=50) | UniPC | 1× | 948 s | - | - | - | 83.09% |
| Ours (T=50) | UniPC | 2.95× | 321 s | 0.178 | 22.72 | 0.748 | 82.62% |
| Wan-14B (T=40) | DPM++ | 1× | 766 s | - | - | - | 82.86% |
| Ours (T=40) | DPM++ | 2.68× | 286 s | 0.169 | 23.00 | 0.765 | 82.45% |

## C   ANALYSIS OF THE RESCALING STRATEGY

We replaced the small model with a simple rescaling strategy similar to CFG-Zero* (Fan et al., 2025), and the results are presented in Table 3. The results indicate that our DMD strategy, which leverages a lightweight neural network for magnitude-aware rescaling, achieves superior performance. In contrast, CFG-Zero*'s rescaling is designed to address inaccuracies in the estimated velocity within CFG and is not effective for cross-step magnitude rescaling. Nonetheless, this approach is promising and warrants further exploration as a potential solution to eliminate dependence on small models.

## D   SCALING TO 720P RESOLUTION

Given the substantial computational overhead associated with high-resolution video generation, we evaluate the scalability of our method on Wan 2.1 using a randomly selected subset of 200 prompts. Table 7 shows that our method scales effectively to higher resolutions.

Table 6: Comparison of rescaling strategies on Wan2.1. We keep the lightweight substitution steps fixed and vary only the rescaling strategy.

| Methods | LPIPS↓ | SSIM↑ | PSNR↑ | VBench↑ |
|---|---|---|---|---|
| w/o rescaling | 0.197 | 21.40 | 0.689 | 81.16% |
| w/ simple rescaling | 0.268 | 20.36 | 0.656 | 80.90% |
| w/o rescaling | 0.178 | 22.72 | 0.748 | 82.62% |

Table 7: Results for Wan 720p, with latency measured under an 8×NVIDIA A100 GPU setup.

| Interval | Efficiency | | Visual Quality | | |
|---|---|---|---|---|---|
| | Speedup↑ | Latency↓ | LPIPS↓ | SSIM↑ | PSNR↑ |
| Wan2.1-14B | 1× | 476 s | - | - | - |
| TeaCache (Liu et al., 2025b) | 2.37× | 201 s | 0.367 | 17.54 | 0.607 |
| SRDiffusion (Cheng et al., 2025) | 2.47× | 193 s | 0.381 | 16.75 | 0.585 |
| Ours ($\tau = 0.002$) | 2.50× | 190 s | 0.276 | 18.93 | 0.661 |
| Ours ($\tau = 0.005$) | 2.77× | 172 s | 0.292 | 18.48 | 0.655 |

# E   DETAILED METRICS ON VBENCH

Table 8 reports all VBench metrics for our DMD strategy. The results show that our method provides effective acceleration while maintaining minimal degradation across the evaluated dimensions. However, as the threshold $\tau$ increases, the growing reuse of directional components reduces their reliability, resulting in a decline in both quality and semantic scores. Therefore, further increasing the reuse rate of directional components remains a key challenge for DMD.

Table 8: VBench Score for all dimensions.

| VBench Scores | Wan2.1-14B | Ours($\tau = 0.002$) | Ours($\tau = 0.002$) | TeaCache | SRDiffusion |
|---|---|---|---|---|---|
| **Total score** | 83.09% | 82.85% | 82.62% | 82.41% | 83.01% |
| **Speedup** | 1× | 2.76× | 2.95× | 2.62× | 2.75× |
| **Quality score** | 85.60% | 85.38% | 85.21% | 84.90% | 85.58% |
| Subject consist | 93.31% | 93.51% | 93.61% | 93.10% | 93.34% |
| Background consist | 96.72% | 96.70% | 96.21% | 95.88% | 96.76% |
| Temporal flickering | 95.98% | 95.50% | 95.63% | 95.79% | 95.95% |
| Motion smoothness | 92.90% | 92.59% | 92.45% | 93.14% | 92.73% |
| Dynamic degree | 43.75% | 43.75% | 43.55% | 43.05% | 44.45% |
| Aesthetic quality | 65.97% | 65.51% | 65.20% | 64.13% | 65.39% |
| Imaging quality | 67.76% | 67.45% | 67.21% | 66.77% | 67.68% |
| **Semantic score** | 73.05% | 72.73% | 72.28% | 72.74% | 72.71% |
| Object class | 85.60% | 85.37% | 84.71% | 85.28% | 85.21% |
| Multiple objects | 69.82% | 69.07% | 68.75% | 67.00% | 68.75% |
| Human action | 82.00% | 82.00% | 82.50% | 83.00% | 82.00% |
| Color | 87.89% | 87.78% | 87.56% | 86.92% | 88.01% |
| Spatial relationship | 81.97% | 82.03% | 81.71% | 81.94% | 81.96% |
| Scene | 38.28% | 38.30% | 36.26% | 36.41% | 39.94% |
| Appearance style | 76.32% | 75.86% | 75.12% | 76.46% | 74.39% |
| Temporal style | 66.48% | 65.58% | 65.33% | 65.82% | 65.85% |
| Overall consist | 69.09% | 68.63% | 68.54% | 68.93% | 68.30% |

# F   MORE VISUALIZATION RESULTS

This section presents additional visual examples of the video generation results in Figures 5 and 6. Video comparisons are available at `https://anonymous.4open.science/r/DMD/`.

*Prompt:"Cute and happy Corgi, a small dog with floppy ears, wagging tail, and sparkling eyes, playing joyfully in a lush green park at sunset. The dog's coat is soft and fluffy, with a mix of black and tan fur tones. It runs and chases after a colorful Frisbee, leaping into the air, surrounded by playful children and families. The park is filled with vibrant flowers, towering trees, and colorful umbrellas. The sun sets behind a cluster of mountains, casting a warm golden glow over everything. Soft, dreamlike lighting creates a surreal atmosphere, blending reality with fantasy. The scene captures the innocence and happiness of the moment, with a sense of magical wonder. The background features ethereal clouds and a starry night sky emerging from the horizon. Surrealism style, medium shot focusing on the dog and its joyful expression."*

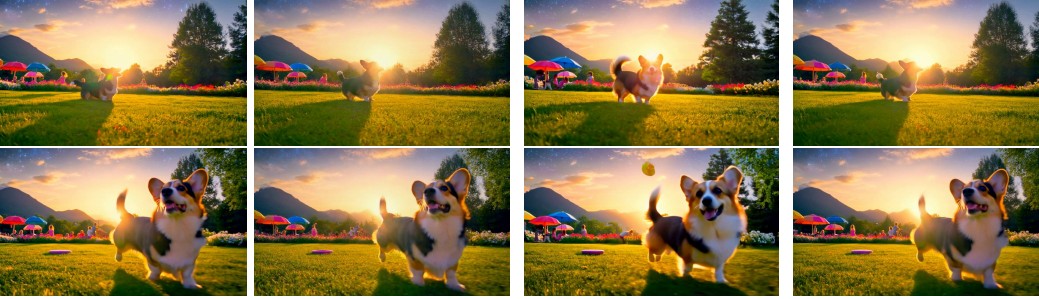

*Prompt:"An astronaut flying in space, depicted in an oil painting style. The astronaut wears a sleek, black spacesuit with reflective patches, floating gracefully in zero gravity. Their helmet glints in the starlight, and their face is illuminated by a soft, golden glow. The background is a vivid, swirling cosmos filled with shimmering nebulae and distant planets. The oil paint technique adds rich textures and vibrant colors, capturing the awe-inspiring beauty of outer space. The astronaut gazes intently at the viewer, a sense of wonder etched on their features. The composition includes a low-angle shot, emphasizing the astronaut's isolation in the vast expanse of space. Oil painting texture with dramatic lighting effects. Low-angle, medium shot focusing on the astronaut's face."*

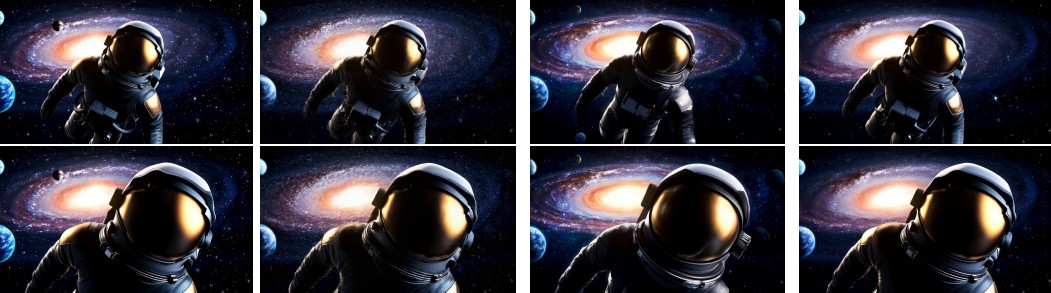

*Prompt:"A joyful fuzzy panda playing a small electric guitar near a cozy campfire. The panda has bright green fur and twinkling blue eyes, surrounded by a warm glow from the flames. In the background, majestic snow mountains with soft, fluffy clouds contrast against a clear, starry night sky. The campfire creates a soft, flickering light, casting shadows on the snow-covered ground. The panda is positioned between the fire and the mountains, its arms gracefully moving as it strums the guitar. The scene is captured with a low-angle shot, emphasizing the serene atmosphere and the lively interaction between the panda and the campfire. Snowflakes gently fall, adding a touch of winter magic. Warm ambient lighting enhances the overall mood."*

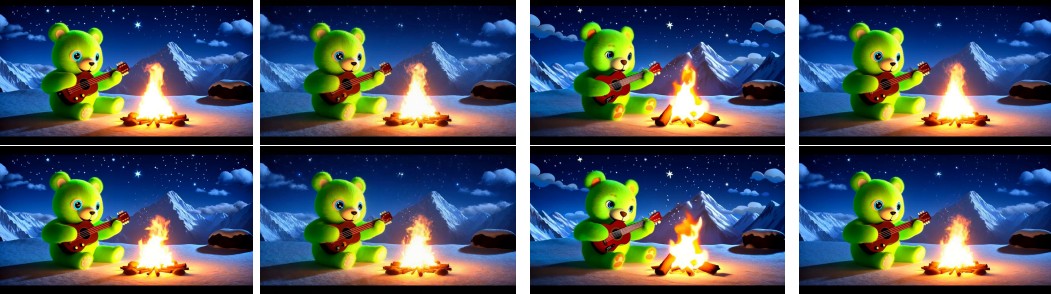

*Prompt:"A tranquil tableau of a delicate porcelain bowl, resting on a polished wooden table in a serene Japanese-inspired living room. The bowl is intricately designed with gold accents and intricate patterns, filled with a few floating lotus flowers in a clear, clear water. The lotus flowers sway gently in the soft, warm sunlight streaming through the large window. The room is adorned with traditional Japanese paper lanterns and hanging scrolls depicting nature scenes. Soft cushions and a tatami mat add to the calming atmosphere. A small tea set sits nearby, ready for a moment of tranquility. The background features a subtle gradient from light beige to a darker shade, highlighting the elegance of the setting. The scene captures a peaceful moment, with slight movements of the lotus leaves and the gentle breeze. Gentle and flowing camera movement, focusing on the bowl and the surrounding elements."*

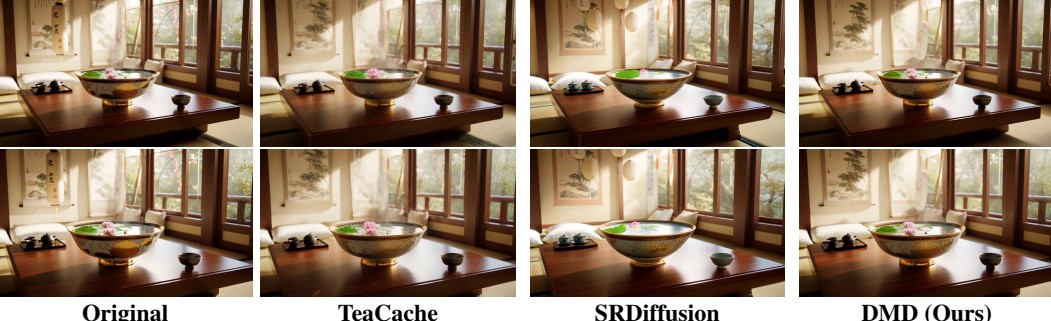

| **Original** | **TeaCache** | **SRDiffusion** | **DMD (Ours)** |

Figure 5: Visualization results on augmented VBench prompts.

*Prompt:"A detailed wooden toy ship with intricately carved masts and sails is seen gliding smoothly over a plush, blue carpet that mimics the waves of the sea. The ship's hull is painted a rich brown, with tiny windows. The carpet, soft and textured, provides a perfect backdrop, resembling an oceanic expanse. Surrounding the ship are various other toys and children's items, hinting at a playful environment. The scene captures the innocence and imagination of childhood, with the toy ship's journey symbolizing endless adventures in a whimsical, indoor setting."*

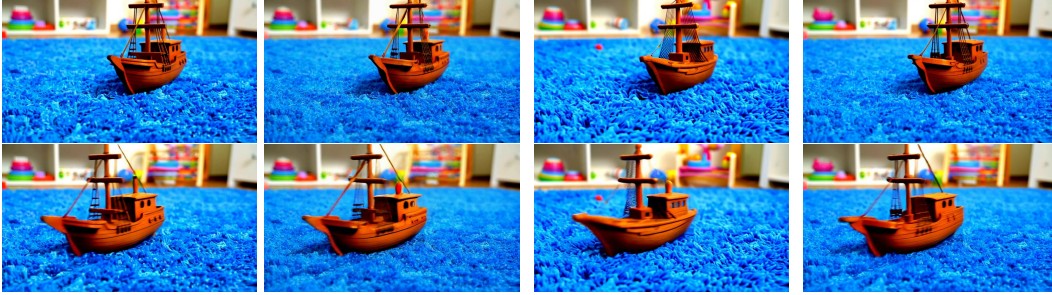

*Prompt:"Hold the camera and enter the warm cafe.comfortable brown leather sofas and log-colored tables and chairs are scattered. the abstract art paintings on the walls are rich in color.the air is filled with the strong aroma of coffee, and the busy figure of the bean grinder and the barista can be seen in the background.the soft light creates a peaceful and comfortable atmosphere. documentary photography style, close-range dynamic perspective."*

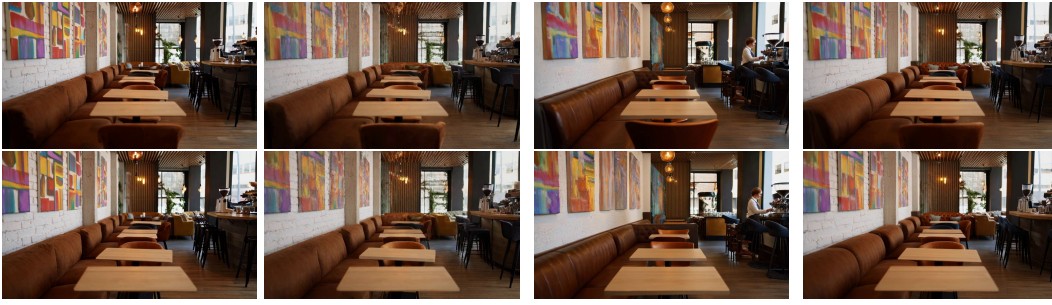

*Prompt:"An elderly gentleman, with a serene expression, sits at the water's edge, a steaming cup of tea by his side. He is engrossed in his artwork, brush in hand, as he renders an oil painting on a canvas that's propped up against a small, weathered table. The sea breeze whispers through his silver hair, gently billowing his loose-fitting white shirt, while the salty air adds an intangible element to his masterpiece in progress. The scene is one of tranquility and inspiration, with the artist's canvas capturing the vibrant hues of the setting sun reflecting off the tranquil sea."*

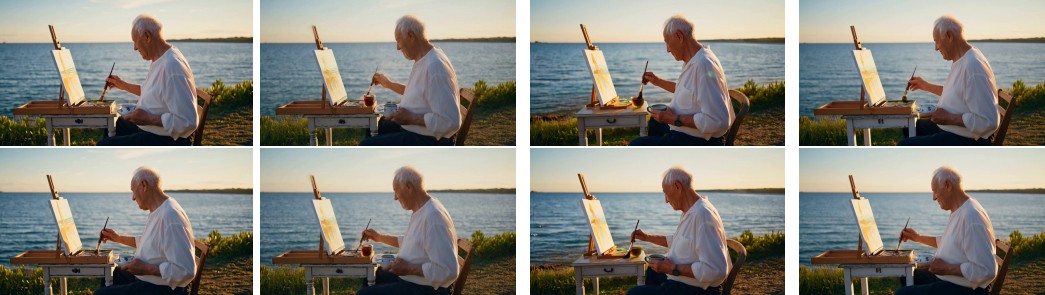

*Prompt:"Furry pink boat,floating on a colorful river of liquid metal. there are all kinds of strange plants and flowers growing in the river, which are colorful and different in shape.the hull is soft and delicate, as if it were made of cotton candy. the background is a fantastic liquid metal landscape, and the light refracts brilliant light.the overall picture is full of fantasy and romantic atmosphere, surrealist style."*

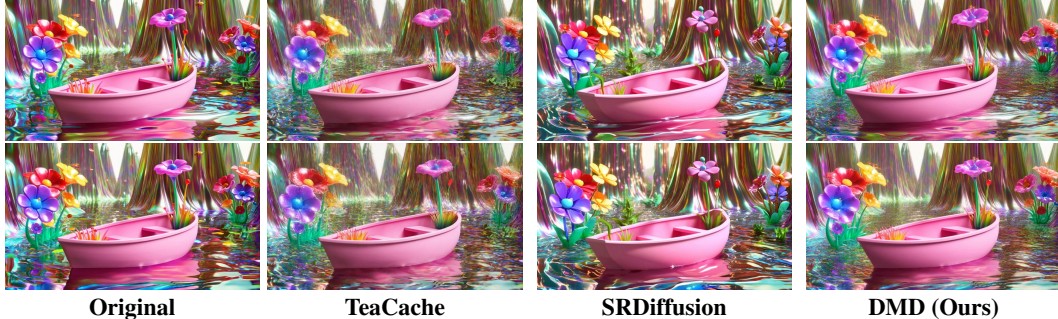

| **Original** | **TeaCache** | **SRDiffusion** | **DMD (Ours)** |

Figure 6: Visualization results on challenging prompts.

