# OpenReview forum: "Direction-Magnitude Decoupling for Fast Video Generation with Flow Matching Models"
_ICLR.cc/2026/Conference — Submitted to ICLR 2026_

### Official Review · Reviewer_YbTF · 2025-10-16

**Soundness:** 3
**Presentation:** 3
**Contribution:** 3
**Rating:** 6
**Confidence:** 3

**Summary:**

This paper proposes a video diffusion acceleration method. The authors observe that in existing caching-based acceleration methods, the generated flow trajectory can deviate from original one due to differences in the magnitude component. To address this issue, they propose modeling the scalar magnitude with a lightweight model, while caching the original diffusion model outputs only for the direction component. Additionally, a CFG variant is introduced to reuse the magnitude information. The proposed method is evaluated on Wan 2.1, demonstrating a 2.95x speedup without much performance degradation.

**Strengths:**

1. The proposed method is well motivated. The authors show an empirical analysis that the error in caching-based methods mainly stem from the magnitude component, which which motivates the proposed magnitude-direction decoupling method.
2. Extensive experimental results demonstrate that the proposed model improves inference speed without much performance degradation.

**Weaknesses:**

1. Lack of user study. The main experiments show improvement is LPIPS, PSNR, SSIM, while the VBench score degrades. This discrepancy raises concern about the actual visual quality. The authors could analyze the VBench results in detail and conduct additional user study to validate both the advantages and limitations of the proposed method.
2. Limited applicability. The proposed acceleration method is restricts to video models that are released with a smaller variant. However, many existing models like HunyuanVideo and Wan2.2-A14B do not provide comparable smaller versions. The authors should discuss the transferability of their approach across different models to address this concern.
3. Lack of experiments at higher resolution. Since the main time overhead of Wan2.1 14B arises from high-resolution inference (>10 min for a 720p video), it would be more informative to evaluate whether the proposed method remains effective at higher resolutions. Such experiments would also help verify the generality of the empirical observations and the robustness of the involved hyperparameters.

**Questions:**

1. The proposed CFG reuse strategy seems related to the velocity rescaling in CFG-Zero* [1] which demonstrates performance gains on its own. Would a simpler velocity normalization like CFG-Zero*, rather than rescaling by a small model's prediction, yield better performance?
2. Small models typically produce motions that are smaller and more subtle. Would approximating the magnitude of a larger model's velocity using the small model's outputs affect the motion intensity in the generated videos?

---

[1] Fan, et al. CFG-Zero⋆: Improved classifier-free guidance for flow matching models. arXiv:2503.18886

---

> ### Author Response · Authors · 2025-11-20
> **Part 1**
>
> Thank you for your positive assessment and helpful feedback. We will revise the manuscript accordingly.
>
> **User study:**
> In the original paper, we provided an anonymous link enabling users to compare multiple baselines via a series of visualized videos. We will also revise the manuscript to include additional examples to broaden the scope of the study.
>
> **Detailed metrics on VBench:**
> Table 1 reports all VBench metrics for our DMD strategy. The results show that our method provides effective acceleration while maintaining minimal degradation across the evaluated dimensions. However, as the threshold $\tau$ increases, the growing reuse of directional components reduces their reliability, resulting in a decline in both quality and semantic scores. Therefore, further increasing the reuse rate of directional components remains a key challenge for DMD.
>
> Table 1. VBench scores for all dimensions (Wan model).
> |VBench Scores|Wan2.1-14B|Ours ($\tau=0.002$)|Ours ($\tau=0.005$)|Teacache|SRDiffusion|
> |-|-|-|-|-|-|
> |**total score**|83.09%|82.85%|82.62%|82.41%|83.01%|
> |**speedup**|1×|2.76×|2.95×|2.62×|2.75×|
> |**quality score**|85.60%|85.38%|85.21%|84.90%|85.58%|
> |subject consist|93.31%|93.51%|93.61%|93.10%|93.34%|
> |background consist|96.72%|96.70%|96.21%|95.88%|96.76%|
> |temporal flickering|95.98%|95.50%|95.63%|95.79%|95.95%|
> |motion smoothness|92.90%|92.59%|92.45%|93.14%|92.73%|
> |dynamic degree|43.75%|43.75%|43.55%|43.05%|44.45%|
> |aesthetic quality|65.97%|65.51%|65.20%|64.13%|65.39%|
> |imaging quality|67.76%|67.45%|67.21%|66.77%|67.68%|
> |**semantic score**|73.05%|72.73%|72.28%|72.74%|72.71%|
> |object class|85.60%|85.37%|84.71%|85.28%|85.21%|
> |multiple objects|69.82%|69.07%|68.75%|67.00%|68.75%|
> |human action|82.00%|82.00%|82.50%|83.00%|82.00%|
> |color|87.89%|87.78%|87.56%|86.92%|88.01%|
> |spatial relationship|81.97%|82.03%|81.71%|81.94%|81.96%|
> |scene|38.28%|38.30%|36.26%|36.41%|39.94%|
> |appearance style|76.32%|75.86%|75.12%|76.46%|74.39%|
> |temporal style|66.48%|65.58%|65.33%|65.82%|65.85%|
> |overall consist|69.09%|68.63%|68.54%|68.93%|68.30%|
>
> **Limited applicability:**
> Our proposed strategy is generally applicable, as residual reuse in flow-matching models primarily arises from errors in the magnitude component, which is also supported by related literature [1]. As a training-free acceleration strategy, our method relies on a lightweight model from the same family, which serves as a prerequisite for the large-small model collaborative inference paradigm [2]. In the future, we believe that an increasing number of model families will offer lightweight versions as options, following the trend already seen in the LLM field, which will help overcome this limitation.
>
> **Scaling to 720p resolution:**
> Due to the high computational cost of high-resolution video generation, we conducted experiments on wan2.1 using a random subset of 200 prompts. Table 2 shows that our method scale effectively to higher resolutions.
>
> Table 2. Results for Wan 720p, with latency measured under an 8×NVIDIA A100 GPU setup.
> |Methods|Speedup↑|Latency↓|LPIPS↓|PSNR↑|SSIM↑|
> |-|-|-|-|-|-|
> |Wan2.1-14B|1×|476 s|-|-|-|
> |Teacache|2.37×|201 s|0.367|17.54|0.607|
> |SRDiffusion|2.47×|193 s|0.381|16.75|0.585|
> |Ours ($\tau$=0.002)|2.50×|190 s|0.276|18.93|0.661|
> |Ours ($\tau$=0.005)|2.77×|172 s|0.292|18.48|0.655|
>
> **Rescaling strategy w/o small model:**
> Thank you for your insightful comment. We replaced the small model with a simple rescaling strategy similar to CFG-Zero*, and the results are presented in Table 3. The results indicate that our DMD strategy, which leverages a lightweight neural network for magnitude-aware rescaling, achieves superior performance. In contrast, CFG-Zero*’s rescaling is designed to address inaccuracies in the estimated velocity within CFG and is not effective for cross-step magnitude rescaling. Nonetheless, this approach is promising and warrants further exploration as a potential solution to eliminate dependence on small models.
>
> Table 3. Comparison of rescaling strategies on Wan2.1. We keep the lightweight substitution steps fixed and vary only the rescaling strategy.
> |Methods|LPIPS↓|PSNR↑|SSIM↑|VBench↑|
> |-|-|-|-|-|
> |w/o rescaling|0.197|21.40|0.689|81.16%|
> |w/ simple rescaling|0.268|20.36|0.656|80.90%|
> |Ours|0.178|22.72|0.748|82.62%|

---

> ### Author Response · Authors · 2025-11-20
> **Part 2**
>
> **Impact on motion intensity:**
> The overall motion pattern is largely determined during the early denoising stages [2], where we still rely on the original large model (Similar to prior work [1,3,4], we preserve the first 20% of the diffusion steps using the large model). Moreover, in the DMD strategy, the lightweight model provides only the magnitude component for calibration rather than serving as a full substitute, which limits its influence. This is further supported by the VBench results in Table 1, where the dynamic degree metric shows no notable degradation, empirically confirming that motion intensity remains largely unaffected.
>
> **References:**
>
> [1] Ma, et al. MagCache: Fast Video Generation with Magnitude-Aware Cache. NeurIPS 2025
>
> [2] Cheng, et al. SRDiffusion: Accelerate Video Diffusion Inference via Sketching-Rendering Cooperation. ArXiv:2505.19151
>
> [3] Liu, et al. Timestep Embedding Tells: It's Time to Cache for Video Diffusion Model. CVPR 2025
>
> [4] Lv, et al. FasterCache: Training-Free Video Diffusion Model Acceleration with High Quality. ICLR 2025

---

### Official Review · Reviewer_uJ31 · 2025-11-01

**Soundness:** 3
**Presentation:** 3
**Contribution:** 3
**Rating:** 8
**Confidence:** 4

**Summary:**

This paper addresses the high computational overhead of flow matching models for video generation by proposing Direction-Magnitude Decoupling (DMD), a training-free acceleration method. DMD leverages two key empirical insights: (1) residual reuse reliably captures the directional component of the original large model’s output, and (2) lightweight small models accurately estimate the magnitude component. It combines these to approximate the original denoising trajectory, adds classifier-free guidance (CFG) reuse to halve small-model inference cost, and uses adaptive switching to mitigate directional error accumulation. Experiments on Wan2.1 and EasyAnimateV5.1 show DMD outperforms baselines (TeaCache, SRDiffusion) with up to 2.95× speedup on Wan2.1 while preserving visual fidelity (e.g., LPIPS 0.178, PSNR 22.72).

**Strengths:**

1. Targeted Empirical Insight: The decoupling of direction/magnitude components is well-motivated and validated (via cosine similarity, ℓ₂ norms), avoiding generic "lightweight substitution" and enabling principled acceleration.

2. Practicality: As a training-free method, DMD requires no retraining or extra data—critical for real-world adoption. CFG reuse is a clever optimization that reduces small-model cost without sacrificing quality.

3. Rigorous Evaluation: Experiments span two base models (Wan2.1, EasyAnimateV5.1) with multiple scales, and use comprehensive metrics (latency, LPIPS, PSNR, SSIM, VBench) to compare against relevant baselines. Multi-GPU results further demonstrate scalability.

4. Error Mitigation: The adaptive switching mechanism (via cumulative directional error) addresses residual reuse’s inherent error accumulation, a key limitation of cache-based methods like TeaCache.

**Weaknesses:**

1. Though the paper positions DMD as a training-free method (avoiding costly retraining or extra data, ), it relies on manual threshold adjustment to balance speed and visual quality—undermining its broad adaptability.

2. The paper acknowledges multiple acceleration paradigms in related work, such as efficient ODE/SDE solvers (e.g., DPM-solver, ), post-training quantization, and progressive distillation. However, it only validates DMD as a standalone method, with experiments limited to comparisons against cache-based (TeaCache) and large-small model collaborative (SRDiffusion) baselines. There is no exploration of combining DMD with other acceleration categories—for example, whether integrating DMD with DPM-solver.

**Questions:**

1. Given that this manual tuning limits DMD’s generalization across different flow matching models or video generation scenarios, does the authors have any preliminary ideas or plans to design an adaptive mechanism (e.g., learning-based threshold adjustment using lightweight signals) that allows DMD to automatically determine the optimal \(\tau\) without manual intervention for new models?

2. Would pairing DMD with quantization distort the model’s ability to accurately estimate direction components, thereby affecting DMD’s performance?

---

> ### Author Response · Authors · 2025-11-20
>
> Thank you for your positive assessment and helpful feedback. We will revise the manuscript accordingly.
>
> **Automated threshold determination methods:**
> Automatically determining the optimal $\tau$ without manual intervention for new models is challenging. Since $\tau$ inherently represents a trade-off between acceleration and quality, completely eliminating manual intervention would reduce flexibility. In practice, a feasible approach is to perform a parameter search over $\tau$ on a small subset of samples from the new model, relating the threshold to perceptual quality metrics. For each candidate value, quality-preservation metrics such as SSIM or LPIPS can be evaluated against the baseline (non-accelerated) results. The largest $\tau$ that maintains an acceptable quality level (e.g., SSIM > 0.7) can then be selected as the final choice.
>
> **Combining DMD with efficient ODE solvers:**
> Efficient ODE solvers aim to achieve high-quality sampling with fewer steps, whereas our method reduces inference cost by replacing parts of the original large-model inference process with lightweight substitutes. Table 1 presents a supplementary analysis of step reduction using DMD with the DPM++ solver, demonstrating its compatibility with efficient ODE solvers.
>
> Table 1. Results of step reduction using DMD combined with efficient ODE solvers.
> |Methods|ODE solver|Speedup↑|Latency↓|LPIPS↓|PSNR↑|SSIM↑|VBench↑|
> |-|-|-|-|-|-|-|-|
> |Wan2.1-14B (T=50)|UniPC|1×|948 s|-|-|-|83.09%|
> |Ours (T=50)|UniPC|2.95×|321 s|0.178|22.72|0.748|82.62%|
> |Wan2.1-14B (T=40)|DPM++|1×|766 s|-|-|-|82.86%|
> |Ours (T=40)|DPM++|2.68×|286 s|0.169|23.00|0.765|82.45%|
>
> **Would pairing DMD with quantization affect DMD’s performance?**
> In DMD, directional components are estimated from the residuals of previous-step predictions, i.e., $v_\theta^\text{est}(x_{t-1}, t-1) = x_{t-1} + r$, with $r = v_\theta(x_t, t) - x_t$. Quantization distortion directly affects $v_\theta(x_t, t)$, reducing the reliability of $v_\theta^\text{est}(x_{t-1}, t-1)$. Therefore, the negative impact on the performance of our DMD is expected to be related to quantization error.
>
> **References:**
>
> [1] Cheng, et al. SRDiffusion: Accelerate Video Diffusion Inference via Sketching-Rendering Cooperation. ArXiv:2505.19151

---

> > ### Comment · Reviewer_uJ31 · 2025-11-20
> >
> > Thank you for your reply, which has clearly resolved my question. For now, I've decided to keep my score.

---

> > > ### Author Response · Authors · 2025-11-26
> > >
> > > Thank you for your effort and for your thoughtful review!

---

### Official Review · Reviewer_heMi · 2025-11-01

**Soundness:** 3
**Presentation:** 3
**Contribution:** 4
**Rating:** 4
**Confidence:** 3

**Summary:**

This paper proposes a training-free acceleration method for flow matching video generation models, called DMD (Direction-Amplitude Decoupling). The core idea is to leverage empirical insights to exploit the complementary advantages between small models and cache residuals: small models within the same family can reliably capture the amplitude information from the original large model's output, while cache residuals can provide precise directional guidance. DMD combines the amplitude estimation from small models with directional calibration guided by cache residuals, adaptively replacing part of the denoising steps. It also introduces CFG (Classifier-Free Guidance) amplitude reuse to further reduce computational overhead, and mitigates error accumulation through an accumulated directional error threshold mechanism.

**Strengths:**

1. Efficiency of CFG reuse and adaptive strategy: Taking advantage of the characteristic of CFG mechanism where conditional and unconditional output amplitudes are similar, an amplitude reuse strategy is proposed to halve the small model inference overhead; meanwhile, an adaptive switching mechanism based on accumulated directional error is designed, which allows for moderate error accumulation while timely invoking the large model for calibration, effectively balancing inference speed and visual quality, and avoiding the error amplification caused by static caching or simple replacement of small models.

2. Practical value of training-free plug-and-play: DMD requires no additional training or fine-tuning, does not rely on large-scale annotated data, and can be directly integrated into existing flow matching models, compatible with multi-scale model architectures (shared VAE), with low deployment cost, possessing strong engineering compatibility and potential for promotion, solving the high cost problem of training-based acceleration methods (such as distillation, quantization).

3. Sufficiency and persuasiveness of experimental validation: The paper conducted comprehensive experiments on two mainstream flow matching models, compared with caching-based and collaborative small-large model baselines, and verified the effectiveness of the method through quantitative metrics (speedup, LPIPS, etc.) and qualitative visualization; meanwhile, it supplemented threshold sensitivity analysis, CFG reuse ablation experiments, and multi-GPU scenario testing, with rigorous experimental design.

**Weaknesses:**

1. The experiments are validated on two models, Wan2.1 and EasyAnimateV5.1. I believe the results on more other mainstream flow matching video generation models (such as HunyuanVideo, Open-Sora) would further enhance the persuasiveness;

2. Detailed metrics on VBench would further enhance the persuasiveness.

3. The direction error accumulation threshold (τ=0.005) in the adaptive strategy is entirely based on empirical settings, without providing theoretical basis or adaptive optimization methods (such as dynamically adjusting the threshold according to the dynamic characteristics of the video), raising doubts about its generality across different scenarios.

4. Can DMD accelerate smaller-sized models?

**Questions:**

Please refer to the weaknesses.

---

> ### Author Response · Authors · 2025-11-20
>
> Thank you for your careful review and valuable comments. We will revise the manuscript accordingly.
>
> **Generality:**
> Thank you for your suggestion. Our proposed strategy is generally applicable, as residual reuse in flow-matching models primarily arises from errors in the magnitude component, which is also supported by related literature [1]. However, models such as HunyuanVideo and Open-Sora do not provide lightweight variants, which are a prerequisite for the large–small model collaborative paradigm [2]. We believe that, in the future, an increasing number of model families will offer lightweight versions as options, following the trend already observed in the LLM field, which will help overcome this limitation of this paradigm.
>
> **Detailed metrics on VBench:**
> Table 1 reports all VBench metrics for our DMD strategy. The results show that our method provides effective acceleration while maintaining minimal degradation across the evaluated dimensions. However, as the threshold $\tau$ increases, the growing reuse of directional components reduces their reliability, resulting in a decline in both quality and semantic scores.
>
> Table 1. VBench scores for all dimensions (Wan model).
> |VBench Scores|Wan2.1-14B|Ours ($\tau=0.002$)|Ours ($\tau=0.005$)|Teacache|SRDiffusion|
> |-|-|-|-|-|-|
> |**total score**|83.09%|82.85%|82.62%|82.41%|83.01%|
> |**speedup**|1×|2.76×|2.95×|2.62×|2.75×|
> |**quality score**|85.60%|85.38%|85.21%|84.90%|85.58%|
> |subject consist|93.31%|93.51%|93.61%|93.10%|93.34%|
> |background consist|96.72%|96.70%|96.21%|95.88%|96.76%|
> |temporal flickering|95.98%|95.50%|95.63%|95.79%|95.95%|
> |motion smoothness|92.90%|92.59%|92.45%|93.14%|92.73%|
> |dynamic degree|43.75%|43.75%|43.55%|43.05%|44.45%|
> |aesthetic quality|65.97%|65.51%|65.20%|64.13%|65.39%|
> |imaging quality|67.76%|67.45%|67.21%|66.77%|67.68%|
> |**semantic score**|73.05%|72.73%|72.28%|72.74%|72.71%|
> |object class|85.60%|85.37%|84.71%|85.28%|85.21%|
> |multiple objects|69.82%|69.07%|68.75%|67.00%|68.75%|
> |human action|82.00%|82.00%|82.50%|83.00%|82.00%|
> |color|87.89%|87.78%|87.56%|86.92%|88.01%|
> |spatial relationship|81.97%|82.03%|81.71%|81.94%|81.96%|
> |scene|38.28%|38.30%|36.26%|36.41%|39.94%|
> |appearance style|76.32%|75.86%|75.12%|76.46%|74.39%|
> |temporal style|66.48%|65.58%|65.33%|65.82%|65.85%|
> |overall consist|69.09%|68.63%|68.54%|68.93%|68.30%|
>
> **Analysis of the cumulative error threshold $\tau$:**
> Our strategy for managing error accumulation is guided by a lightweight neural network that is sensitive to sample difficulty. In other words, the network can assess the redundancy level of each sample and output relatively large directional errors for those with low redundancy, thereby accelerating the recalibration process performed by the original large model. Consequently, our design aims to maintain a fixed threshold to ensure that the accumulated error remains below a specified limit, while also enabling sample-adaptive dynamic acceleration across diverse scenarios.
>
> **Can DMD accelerate smaller-sized models?**
> Yes. Consistent with previous large-small model collaborative inference [2]，DMD accelerates a diffusion model by leveraging a lighter counterpart within the same family. Thus, as long as a lighter version exists, DMD can be applied.
>
> **References:**
>
> [1] Ma, et al. MagCache: Fast Video Generation with Magnitude-Aware Cache. NeurIPS 2025
>
> [2] Cheng, et al. SRDiffusion: Accelerate Video Diffusion Inference via Sketching-Rendering Cooperation. ArXiv:2505.19151

---

### Official Review · Reviewer_ikti · 2025-11-01

**Soundness:** 3
**Presentation:** 2
**Contribution:** 3
**Rating:** 4
**Confidence:** 4

**Summary:**

The authors address the significant computational latency in flow matching models for video generation, a well-known bottleneck for iterative sampling. They observe that existing training-free acceleration methods, such as caching mechanisms or small-model substitution, often compromise visual fidelity. The reason, they posit, is that these substitutes can deviate from the original model's denoising trajectory, leading to accumulated errors. The paper's primary investigation involves decoupling the model's velocity field output into its direction and magnitude components. Their central empirical finding is that these two components are best approximated by different lightweight methods: cached residuals (residual reuse) provide a highly reliable estimate for the direction, while lightweight models from the same family robustly capture the magnitude.

Building on this analysis, the authors propose a novel training-free method, Direction-Magnitude Decoupling (DMD). The core of this technique is to construct a more faithful lightweight substitute by synthesizing a new velocity vector. This vector combines the magnitude estimated by the small model with the direction provided by the cached residual. This "direction-calibrated" approach allows the model to stay significantly closer to the original denoising path, thus preserving quality. The framework is made adaptive, using a cumulative error threshold to determine when to briefly invoke the large model for correction. Furthermore, a CFG reuse strategy is introduced to exploit redundancies in classifier-free guidance, further reducing inference costs. The authors report that DMD achieves substantial speedups, such as $2.95\times$ on the Wan2.1 model, while maintaining superior visual fidelity compared to existing acceleration baselines.

**Strengths:**

+ The central insight of decoupling the model output into direction and magnitude components and leveraging the complementary strengths of small models and residual reuse is novel. This  design leads to improved performance.

+ DMD achieves superior performance across all key metrics (Speedup, LPIPS, PSNR, SSIM) on two different model families (Wan2.1 and Easy Animate V5.1). The $2.95\times$ speedup on Wan2.1 while improving LPIPS to $0.178$ is a significant result.

+ As a training-free acceleration method, DMD is accessible and practical for immediate adoption with existing large flow matching models.

+ The introduction of the adaptive strategy using cumulative directional error $\mathcal{E}$ (Equation 5) and the CFG reuse mechanism demonstrates a robust and well-thought-out system design for maintaining fidelity under acceleration.

**Weaknesses:**

The empirical analysis focuses heavily on the $20\%$ to $95\%$ range of the diffusion process (Figure 2). While this covers the bulk of the steps, a brief discussion or analysis of the decoupling behavior in the extreme initial (0% to 20%) and final (95% to 100%) stages would complete the picture. Specifically, the paper mentions preserving the first 20% of steps with the large model but does not fully detail the inherent component reliability in those initial steps. The cumulative error threshold $\tau$ is set empirically, and while the ablation is good, a deeper theoretical justification or guidance for setting $\tau$ based on model architecture or dataset properties would be a valuable addition.

**Questions:**

- The paper mentions the magnitude for conditional output $v_{\varphi}(x_{t},t|c)$ and unconditional output $v_{\varphi}(x_{t},t|c=\emptyset)$ is nearly identical (Figure 2c, 2f). Is the small model used in these plots a different version for conditional vs. unconditional generation, or is it a single model trained to predict both?

- Could the authors provide a brief analysis or discussion of the component reliability (directional vs. magnitude) in the initial $0\%$ to $20\%$ of the diffusion process, which is currently performed by the large model?

- While the speedup is excellent, the comparison in Figure 4 is limited to a small number of GPUs (up to 6). Providing scalability analysis up to $8$ or $16$ GPUs would further strengthen the claim of high efficiency in multi-GPU setups.

---

> ### Author Response · Authors · 2025-11-20
>
> Thank you for your careful review and valuable comments. We will revise the manuscript accordingly.
>
> **Analysis of our DMD strategy in the initial denoising process:**
> In the early denoising stage, the model needs to synthesize semantic content from pure noise [1]. Unlike the intermediate stage (20%–95%), the initial phase is critical and exhibits low redundancy, as confirmed by prior post-hoc acceleration studies [1–3]. Consequently, the directional component, which relies on redundancy-based residual reuse, is constrained at this point. In contrast, magnitude estimation remains relatively stable, as it is derived from lightweight neural networks. Quantitative results demonstrating the effects of applying lightweight substitutions in the early denoising stage are presented in Table 1. The results show that lightweight substitutions in these critical early steps incur substantial performance degradation.
>
> Overall, it is generally advisable to rely on the original large model for reliable generation during the early denoising phase, a strategy that has also been widely adopted in post-hoc acceleration methods [1–3].
>
> Table 1. Empirical analysis of lightweight substitution intervals. The threshold $\tau$ is set to 0.005.
> |Interval|Speedup↑|Latency↓|LPIPS↓|PSNR↑|SSIM↑|VBench↑|
> |-|-|-|-|-|-|-|
> |0%–95%|3.85×|246 s|0.567|13.43|0.431|79.02%|
> |20%–95% (Ours)|2.95×|321 s|0.178|22.72|0.748|82.62%|
>
> **Analysis of the cumulative error threshold $\tau$:**
> Our strategy is based on the architecture-agnostic cosine error of the output velocity vector, and error accumulation is guided by a lightweight neural network that is aware of sample difficulty. Therefore, our strategy is relatively robust to different model architectures and dataset characteristics. Our ablation study shows that a threshold of 0.002 strikes a good balance between quality and efficiency, though lowering the threshold can be considered if higher fidelity is desired.
>
> **Clarification regarding Figures 2c and 2f:** The small model used in these plots is a unified Classifier-Free Guidance (CFG)-trained model capable of performing both conditional and unconditional predictions.
>
> **Scaling to more GPUs:**
> Table 2 provides supplementary results on scaling DMD to 8 GPUs, demonstrating its effectiveness in multi-GPU configurations.
>
> Table 2. Multi-GPU inference efficiency analysis.
> |GPUs|TeaCache|SRDiffusion|Ours|
> |-|-|-|-|
> |1|2.62×|2.76×|2.95×|
> |2|5.12×|5.48×|5.78×|
> |4|9.77×|10.42×|11.15×|
> |6|14.36×|15.05×|16.07×|
> |8|16.50×|17.44×|18.58×|
>
> **References:**
>
> [1] Cheng, et al. SRDiffusion: Accelerate Video Diffusion Inference via Sketching-Rendering Cooperation. ArXiv:2505.19151
>
> [2] Liu, et al. Timestep Embedding Tells: It's Time to Cache for Video Diffusion Model. CVPR 2025
>
> [3] Lv, et al. FasterCache: Training-Free Video Diffusion Model Acceleration with High Quality. ICLR 2025

---

### Author Response · Authors · 2025-12-03
**Rebuttal Summary**

We thank the reviewers for their thorough and constructive comments. Our key contributions are as follows (from the reviews):

* **Core Insight & Novelty:** Reviewers (ikti, YbTF, uJ31) found the problem well-motivated and the central idea of decoupling direction and magnitude components to be a novel approach to acceleration.

* **Methodological Design:** Our key empirical finding is that direction and magnitude are best approximated by different lightweight strategies: cached residuals reliably estimate direction, while small family models capture magnitude. Reviewers praised the robust system design, noting that CFG reuse (halving small-model overhead) and adaptive switching based on cumulative directional error effectively balance speed and visual quality (ikti, heMi, uJ31).

* **Experimental Validation**: Reviewers (ikti, uJ31) acknowledged the superior performance across all key metrics (Speedup, LPIPS, PSNR, SSIM) on two model families. The $2.95\times$ speedup on Wan2.1 while maintaining fidelity was cited as a significant result. Crucially, DMD was recognized as a practical, training-free plug-and-play method, essential for adoption.

Based on the reviewers’ valuable feedback, we have conducted additional experiments and revised the manuscript accordingly, which we hope sufficiently address the reviewers’ concerns. The major additions and improvements are summarized as follows:

* We have clarified in Figures 2c and 2f that both the conditional and unconditional outputs are generated by a single model (ikti(Q1)).

* We have provided an analysis of the DMD strategy used in the initial denoising process in Appendix A (ikti(Q2)).

* We have provided a discussion on the threshold for directional error accumulation (ikti, heMi, uJ31).

* We now include supplementary results on scaling DMD to 8 GPUs (ikti(Q3)).

* We have discussed the generalization capability of the proposed method (heMi(W1, W4), uJ31(Q2), YbTF(W2)).

* We have added an analysis of integrating our method with other efficient ODE solvers in Appendix B (uJ31(W2)).

* We have included an analysis of the rescaling strategy without using small models in Appendix C (YbTF(Q1)).

* We have added experiments on scaling to 720 solution steps in Appendix D (YbTF(W3)).

* We have provided VBench scores for all dimensions in Appendix E (YbTF(W1), heMi(W2)).

* We have included additional visualizations in Appendix F, including augmented VBench prompts and challenging prompts, and have updated the anonymized link with more video visualizations (YbTF(W1)).

* We have added an analysis of the impact of DMD on motion intensity (YbTF(Q2)).

Although OpenReview experienced a technical issue (on Nov. 27), prior to the outage (on Nov. 20), Reviewer uJ31 had acknowledged and affirmed our rebuttal and decided to retain a positive evaluation (score 8: accept).

We thank all reviewers and Area Chairs for you efforts.

Best, Authors

---

### Meta-Review · Area_Chair_Xqiz · 2026-01-07

**Summary:**

Summary of major concerns:

### Reviewer ikti
1. Missing discussion on the behavior of the 0%-20% and 95%-100% stages:
    * **Author replies**: The authors add further clarifications and also the results of applying
    the method in the 0%-20% interval, and the results show that the performance drops significantly.
    * **AC comment**: I think the concern is well addressed.
2. The error accumulation threshold is set empirically, and there is no theoretical justification.
    * **Author replies**: The authors argue that the method is based on architecture-agnostic cosine error
    and that the method is relatively robust to different model architectures and datasets.
    * **AC comment**: I think the conconer remains unresolved. It's unclear whether different prompts
    may require different tuning of the threshold, which will greatly limit the applicability of the method.

### Reviewer heMi
1. The paper only experiments with two models. More models need to be added.
    * **Author replies**: Many open-sourced models don't have lightweight variants and cannot be
    applied here.
    * **AC comment**: I think the concern is well addressed.

2. The error accumulation threshold is set empirically, and there is no theoretical justification.
    * See Reviewer ikti (2)


### Reviewer uJ31
1. Reply on manual threshold adjustment:
    * See Reviewer ikti (2)
2. Whether the proposed method can be combined with other solvers
    * **Author replies**: The authors added additional results with DPM-solver and UniPC and show the method
    works well.
    * **AC comment**: I think the concern is well addressed.


### Reviewer YbTF
1. Lack of user study.
    * **Author replies**: The authors added more video results in an anonymous link.
    * **AC comment**: I think the concern remains unresolved. From the video results, I can see that
    the visual quality between SRDiffusion and the proposed method is similar. Particularly, it makes
    me concerned that it may not be appropriate to use the original sampling results as ground truth and
    use SSIM/PSNR/LPIPS as metrics, considering that it's possible for the method to generate different
    visual results but still have good quality and match the prompt.

**Reviewer Concerns:**

See above.

I think the concerns about the empirical threshold and the evaluation remain unresolved.

**Reviewer Scores:**

Reviewer ikti: maintain the score 4

Reviewer heMi: maintain the score 4

Reviewer uJ31: maintain the score 8

Reviewer YbTF: lower the score from 6 to 4

---

### Decision · Program_Chairs · 2026-01-26

Reject